# Biased Opioid Antagonists as Modulators of Opioid Dependence: Opportunities to Improve Pain Therapy and Opioid Use Management

**DOI:** 10.3390/molecules25184163

**Published:** 2020-09-11

**Authors:** Wolfgang Sadee, John Oberdick, Zaijie Wang

**Affiliations:** 1Department of Cancer Biology and Genetics, College of Medicine, The Ohio State University, Columbus, OH 43210, USA; 2Aether Therapeutics Inc., 4200 Marathon Blvd. Austin, TX 78756, USA; 3Pain and Addiction Research Center, University of California San Francisco, San Francisco, CA 94158, USA; 4Department of Neuroscience, College of Medicine, The Ohio State University Wexner Medical Center, Columbus, OH 43210, USA; John.Oberdick@osumc.edu; 5Departments of Pharmaceutical Sciences and Neurology, University of Illinois at Chicago. Chicago, IL 60612, USA; zjwang@uic.edu

**Keywords:** μ opioid receptor, receptor model, biased ligands, dependence, pain therapy, neonatal opioid withdrawal syndrome, naltrexone, 6β-naltrexol, buprenorphine

## Abstract

Opioid analgesics are effective pain therapeutics but they cause various adverse effects and addiction. For safer pain therapy, biased opioid agonists selectively target distinct μ opioid receptor (MOR) conformations, while the potential of biased opioid antagonists has been neglected. Agonists convert a dormant receptor form (MOR-μ) to a ligand-free active form (MOR-μ*), which mediates MOR signaling. Moreover, MOR-μ converts spontaneously to MOR-μ* (basal signaling). Persistent upregulation of MOR-μ* has been invoked as a hallmark of opioid dependence. Contrasting interactions with both MOR-μ and MOR-μ* can account for distinct pharmacological characteristics of inverse agonists (naltrexone), neutral antagonists (6β-naltrexol), and mixed opioid agonist-antagonists (buprenorphine). Upon binding to MOR-μ*, naltrexone but not 6β-naltrexol suppresses MOR-μ*signaling. Naltrexone blocks opioid analgesia non-competitively at MOR-μ*with high potency, whereas 6β-naltrexol must compete with agonists at MOR-μ, accounting for ~100-fold lower in vivo potency. Buprenorphine’s bell-shaped dose–response curve may also result from opposing effects on MOR-μ and MOR-μ*. In contrast, we find that 6β-naltrexol potently prevents dependence, below doses affecting analgesia or causing withdrawal, possibly binding to MOR conformations relevant to opioid dependence. We propose that 6β-naltrexol is a biased opioid antagonist modulating opioid dependence at low doses, opening novel avenues for opioid pain therapy and use management.

## 1. Introduction

The μ opioid receptor (MOR) is the main target of opioid analgesics, providing strong pain relief but also causing multiple adverse effects and addiction. Documented to exist in multiple forms with distinct functions, MOR and its ligands elicit a perplexingly broad spectrum of effects—opening the opportunity for discovering opioid analgesics with reduced adverse effects. Among these, biased agonist ligands can be directed to stimulate optimal MOR signaling properties [1]. On the other hand, biased MOR antagonists capable of blocking deleterious signaling or regulatory pathways have received less attention. Reviewing documented opioid drug effects, we propose a novel receptor model that can account for diverse pharmacological effects of MOR ligands, including biased antagonists. The type of ligand considered here is thought to differ from allosteric modulators of MOR [2,3] by interacting with the orthosteric site of agonist binding. Biased MOR antagonists could serve as modulators of opioid dependence, for improved pain therapy and opioid use management.

## 2. Evidence for Multiple Receptor Conformations with Distinct Signaling Pathways, and the Potential of Biased Agonists

G protein coupled receptors (GPCRs) are flexible membrane proteins that require lipids, signaling proteins, and co-factors or ligands to attain conformational stability. As a result, GPCRs exist in various conformations as a function of cellular environment, each accepting a distinct spectrum of ligands associated with distinct signaling or regulatory pathways [1,4,5,6]. These properties of GPCRs have triggered a search for biased agonists that selectively activate one pathway over the other, to enhance desired pharmacological outcomes relative to adverse effects [6,7,8,9,10]. MOR indeed appears to exist in multiple forms with distinct signaling and regulatory pathways [1]. Focus in the opioid field has been on creating efficacious pain therapies without unwanted side effects, including tolerance, dependence, and drug craving—hallmarks of addiction—respiratory depression, and various adverse effects (constipation and opioid-induced bowel dysfunction, bone loss, immune dysregulation, nausea, and more) [1,2,3,11]. MOR is considered a main mediator of these actions. Separating desirable from adverse effects is a central goal of current opioid research.

Biased agonists binding to receptors either coupled to G proteins or interacting with other scaffolding proteins including beta-arrestins have taken center stage in the search for opioid analgesics with less tolerance and withdrawal effects [1,12] and low respiratory depression potential [9]—the latter is a cause of countless overdose deaths. While beta-arrestins are thought mainly to orchestrate receptor desensitization, they can also activate tyrosine kinases and downstream signaling [13]. Results with biased opioid agonists are promising but must still be viewed with caution, as none are as of yet approved for general use.

Some reports propose that opioid receptors and specifically MOR exist in different forms in peripheral versus central neurons [13]. Peripheral and central opioid receptor systems could interact dynamically, for example in the induction of opioid induced hyperalgesia, reported to be mediated both centrally and peripherally [14,15]. Peripheral MOR sites could have relevance to inflammation induced neuropathic pain, invoking beta-arestin-2 silenced MOR sites in afferent nociceptors [14,16], that get activated upon inflammatory stimuli. MOR activation could then suppress pain sensation, but also lead to a vicious circle of sustained neuropathic pain [14,16]. While an attractive model to account for the presence of ‘silent‘ MOR sites, the evidence is still missing how the activation occurs and whether these silent MOR sites perform signaling not measured by conventional means.

All extant MOR signaling models can account for only part of the astounding diversity of pharmacological effects observed in countless published studies, including unexpected opioid antagonist effects—the focus of this article to explore opportunities for developing safer opioid analgesics.

## 3. A Basally Active Receptor Mediating MOR Signaling (MOR-μ*)

Capable of signaling spontaneously, GPCRs are restrained in an inactive ground state consisting of large complexes with lipids, proteins, and co-factors. Upon binding to this ground state, agonists trigger a change in receptor conformation sufficient to release constraints keeping the receptor silent, thereby, initiating the signaling cascade. Hence, ground and activated receptor states of the μ opioid receptor, designated here as MOR-μ and MOR-μ*, respectively, assume distinct conformations, often leading to reduced agonist binding affinity to MOR-μ* (Figure 1) [17,18,19]. Typical opioid drugs such as morphine and etorphine potently bind to the ground state MOR-μ but rapidly dissociate from MOR-μ* [20]. However, continued association of the agonist with the activated receptor is also possible and could contribute to biased agonist signaling for some GPCRs [9]. We have shown that the ultra-potent etorphine enters the rat brain and occupies only 1% of MOR sites at its antinociceptive EC50 (<0.001 mg/kg) [20], previously suggested to indicate a ‘receptor reserve’ [21]. It appears more likely that etorphine binds with high affinity to MOR-μ and upon rapid activation dissociates from MOR-μ*, which carries out the signaling process in the absence of agonist ligand. Consistent with this hypothesis, we had shown that the etorphine off-rate has a dissociation half-life (t1/2) less than 1 min in vivo (rat brain), whereas, after sacrifice and tissue homogenization, the dissociation t1/2 increases to ~40 min [20]—the in vivo activation state no longer remains intact. Therefore, it is important to consider the relative affinities and effects of ligands at both MOR-μ and MOR-μ* in life tissues, to understand opioid effects.

We and others had further demonstrated that the ground state MOR-μ receptor can spontaneously convert to active MOR-μ*, in the absence of any ligand (Figure 1) [17,18,19,22], as demonstrated for numerous GPCRs. Moreover, basal MOR-μ* activity increases upon sustained opioid agonist exposure and appears to play a role in opioid dependence [17,18,19,22]—the mechanism by which elevated MOR-μ* signaling is maintained over time remains elusive. Enhanced MOR-μ* activity results in high sensitivity to inverse opioid antagonists such as naloxone and naltrexone, apparently acting at the ligand-free MOR-μ* in a non-competitive fashion, with as little as 50–100 microgram naloxone given iv causing aversive reactions in methadone-managed opioid use patients (typically receiving 50–100 mg/day methadone). We propose that pharmacological MOR antagonist effects reflect binding affinity and efficacy at both MOR-μ and MOR-μ*. Three opioid drugs serve to illustrate these interactions.

*Naltrexone:* Naltrexone is clinically used to prevent opioid relapse and reduce alcohol binge drinking [23,24,25]. An inverse antagonist, naltrexone suppresses basal MOR-μ* activity and thereby potently causes withdrawal symptoms in dependent subjects [19,26,27,28,29]. In addition, naltrexone antagonizes antinociception of 30 mg/kg morphine with an IC50 of 0.007 mg/kg in mice [26] (Table 1). This extraordinary potency against a high agonist dose can be accounted for by non-competitive binding of naltrexone to morphine-generated ligand-free MOR-μ*, thereby suppressing signaling activity. Similar high naltrexone potency has been reported in rhesus monkeys against both fentanyl analgesia and in causing withdrawal in dependent animals (pA2 8.5 mg/kg) [29] (Table 1). Because strong naltrexone-induced withdrawal reactions continue in dependent subjects even after the opioid drug has been fully excreted, naltrexone therapy to prevent relapse is started only 1–2 weeks after complete opioid withdrawal [30].

*6β-naltrexol (6BN):* Naltrexone is converted to its main metabolite 6BN, a neutral antagonist (Figure 2) [19,27,28]. With the hypothesis that 6BN binds potently to MOR-μ* without suppressing signaling, we propose that 6BN blocks opioid analgesia or causes withdrawal only at much higher doses (Table 1) because it needs to compete with the opioid agonist at MOR-μ (Figure 1). Even though in vitro MOR binding affinity is nearly equal to that of naltrexone (Ki 3.2 nM vs. 1.7 nM, respectively, in rhesus monkeys [29]), 6BN is >100-fold less potent than naltrexone in blocking antinociception and causing withdrawal, in mice, guinea pigs, and rhesus monkeys [26,27,28,29] (Table 1). For example 6BN has an ID50 of 1.3 mg/kg in reversing morphine antinociception in mice vs. 0.007 mg/kg naltrexone [26]. In view of near equal binding affinity at MOR (Table 1), these results cannot be fully accounted for by slower access of 6BN to the brain (see below), but are resolved if 6BN indeed binds potently to MOR-μ* while not preventing MOR-μ* signaling, acting as a neutral antagonist. At higher doses only, 6BN is capable of competing with the opioid agonist at MOR-μ, preemptively preventing its activation to MOR-μ* by an agonist. As a result, 6BN blocks opioid analgesia only at high doses, or requires high doses to cause withdrawal in a dependent subject present [18,31]. After withdrawal when the opioid is excreted, for example 24 h after the last morphine dose in mice, 6BN no longer causes withdrawal. In contrast, naloxone and naltrexone still elicit substantial withdrawal at 24 h and later by blocking MOR-μ* activity, which is sustained and thereby also maintains the dependent state [27].

These properties of neutral MOR antagonists such as 6BN, naloxol, and naltrexamine, and their derivatives [31], offer new approaches to the management of drug use disorder. Similar receptor models may also apply to other opioid receptors (DOR and KOR), and more broadly to other GPCR families. 6BN and its analogues bind to MOR, DOR, and KOR, but inverse and neutral efficacy may differ between receptors; for example, 6BN acts as an inverse agonist at KOR after agonist pretreatment in tissue culture, whereas naltrexone appears to be neutral [32]. The influence of 6BN interactions with DOR and KOR remain to be studied.

*Buprenorphine.* For treatment of opioid use disorder, buprenorphine has been adopted broadly as it has only intermediate efficacy at MOR, but suppresses drug craving [33]. Mixed agonists–antagonists are less efficacious analgesics that can cause withdrawal in highly dependent subjects if they fail to elicit the level of MOR signaling needed in profound dependence [34]. Partial opioid agonists and mixed agonists–antagonists may stay engaged with MOR-μ for a longer time period before activation to MOR-μ*, and they also could retain some affinity for MOR-μ*. In animal studies, buprenorphine displays an unusual inverse bell-shaped dose–response curve in antinociceptive tests and in drug seeking behavior [35,36], antagonizing its own action at very high doses. We had observed that administration of high buprenorphine doses to rats first leads to sedation and catatonia, but when more buprenorphine floods into the brain, the animals wake up and behave normally, only to revert to a catatonic state when drug levels begin to decrease again before returning to normal activity (unpublished observations). Several hypotheses have been proposed to account for this pharmacological effect, but the answer has remained delusive. Considering the MOR model shown in Figure 1, a parsimonious solution offers itself: assume buprenorphine activates MOR-μ to an intermediate level, it then dissociates and enables MOR-μ* signaling to occur. However, buprenorphine could have residual and sufficient affinity to MOR-μ* to bind to it when given at higher doses, but then acting as an inverse agonist at MOR-μ*. In this fashion, buprenorphine can indeed antagonize its own action at high doses. While this hypothesis requires further testing, it can serve as a conceptual template for new drug development.

## 4. Peripherally Active μ Opioid Receptor Antagonists (PAMORA) and 6β-Naltrexol

The peripheral opioid system plays multiple roles, for example in the g.i. tract and in nociceptor neurons, the latter involved in peripheral analgesia [37]. PAMORAs including methylnaltrexone, naloxegol, alvimopan, and naldemdine are in clinical use to treat opioid-induced bowel dysfunction and constipation [15,38]. Peripheral selectivity is thought to depend on limited access to the CNS through the blood–brain-barrier (BBB), either because of high polarity or by extrusion via export transporters [39]. Similarly, 6BN has somewhat restricted access to the CNS, in part accounting for its peripheral selectivity. This leads to 5–10 fold higher 6BN blood over brain levels, and higher potency in blocking opioid effects on the g.i. tract in mice compared to centrally mediated opioid antinociception [40]. In opioid-naïve human volunteers, 6BN blocks morphine-induced slowing of bowel movements with an IC50 of ~3 mg (both drugs given i.v.), whereas analgesia measured in a cold pressure assay was unaffected by the highest tested dose of 20 mg 6BN [41].

Recent results indicate that 6BN’s peripheral selectivity is not solely due to slow penetration of the BBB. Whereas, we have found higher blood than brain 6BN levels in mice [42] and guinea pigs [43], 6BN enters the brain of rhesus monkeys with less restriction, resulting in equal blood and brain levels (Figure 3). Yet, 6BN is >100-fold less potent than naltrexone in blocking fentanyl antinociception and in causing withdrawal in rhesus monkeys [29], similar to what is found in mice and guinea pigs. Testing the potency of naltrexone and 6BN in antagonizing the fentanyl-induced suppression of electrically stimulated peristalsis in the guinea pig ileum, Porter et al. [26] reported IC50 concentrations of 0.26 and 0.09 nM, respectively, showing that 6BN was not only highly potent in this assay, but also more potent than naltrexone, in contrast to its slightly lower affinity to MOR measured in vitro. These results are inconsistent with canonical MOR receptor models, but rather suggests the presence of additional MOR conformations of yet unknown structure and function, with high 6BN affinity.

The possible presence of distinct MOR conformations, or varying relative abundance between conformations as a function of cellular environment, and in the periphery compared to the CNS, has been reviewed by Jeske [13], suggesting that beta-arrestin coupled MOR sites in afferent nociceptors account for their silent status, being activated only upon inflammatory stimuli. MOR-μ* basal activity appears also to be absent in peripheral afferent nociceptor neurons, but emerges upon nociceptive stimuli as a physiological countermeasure leading to abatement of neuropathic pain [16,44]. However, if such basal MOR activity fails to be reversed, it can contribute to chronic neuropathic pain. Only 6BN, but not naltrexone, can facilitate the reversal of chronic neuropathic pain, revealing the biological relevance of MOR-μ* basal activity [16,44]. Whether such MOR sites also exist in the g.i. tract remains to be determined. The finding of extreme 6BN potency in the guinea pig ileum indicates the existence of MOR sites at which 6BN may act with high potency in a non-competitive manner.

We have observed that 6BN becomes more potent in blocking gastrointestinal effects of morphine in mice when pre-administered, with maximum potency reached at ~100 min before morphine [26]. This result supports a model in which MOR exists in two different conformations, namely MOR-μ and a novel MOR form of yet unknown function. We hypothesize that this MOR state is in equilibrium with MOR-μ and is stabilized by 6BN binding, shifting the equilibrium away from MOR-μ, thereby preventing activation to MOR-μ*. Such equilibrium between receptor states could exist in the CNS as well, but with tissue specific preference for one form or the other—a potential mechanism for peripheral selectivity of some opioid ligands. This MOR model also predicts novel actions of ligands such as 6BN in modulating MOR signaling—for example affecting opioid dependence.

## 5. 6BN Prevents Development of Opioid Dependence with High Potency

Repeated use of opioid analgesics leads to tolerance, dependence, hyperalgesia, and drug seeking behavior. All these effects underlie distinct processes while common mechanisms may also exist leading to opioid addiction. We have postulated that increased and sustained formation of MOR-μ* characterizes the dependent state, accounting for the high potency of inverse agonists to elicit withdrawal behavior [17,18,19,22]. Here, we address the novel hypothesis that an as yet poorly defined receptor conformation may be involved in dependence, possibly with high affinity for 6BN.

In a first set of experiments, we tested whether 6BN given together with daily doses of morphine (10–20 mg/kg) for 6 days prevents naloxone-induced withdrawal behavior in juvenile mice (5–15 days old)—with the goal of developing a model for preventive therapy for neonatal opioid withdrawal syndrome (NOWS). In juvenile mice, 6BN readily enters the brain as the BBB remains underdeveloped until day 20 post-partum, while naloxone-induced opioid dependence can be readily measured at 10–18 days after birth [42]. Co-administration of 6BN with morphine potently prevents naloxone-induced withdrawal, tested 3 h after the last dose. 6BN displayed an IC50 of ~0.03 mg/kg (Figure 4) [42], substantially below the expected antinociceptive IC50 in adult mice [27]. In this experimental design, morphine is not yet completely eliminated from the circulation at time of testing, yielding a rather shallow dose–response curve as naloxone acts by both blocking MOR-μ* and antagonizing morphine at MOR-μ, the latter process not expected to be affected by 6BN. In addition, we had observed that naloxone-induced withdrawal jumping was delayed even at the lowest dose of 6BN tested (0.0067 mg/kg) [42]. These results suggest that 6BN reduces or prevents dependence at exceedingly low doses that do not block antinociceptive effects nor cause immediate withdrawal.

Encouraged by these results, we subsequently tested co-administration to adult guinea pigs of 6BN with methadone (10 mg/kg) for 3 days, with withdrawal testing on day 4, finding an IC50 6BN dose of ~0.01 mg/kg to block naloxone-induced locomotion [43], two orders of magnitude below the dose required to block antinociception [26]. Similarly, co-administration of an s.c. dose as low as 0.03 mg/kg 6BN completely suppresses naloxone-induced withdrawal jumping in adult mice made dependent on morphine (10 mg/kg for 5 days) (Z. Wang; unpublished data). Lastly, Oberdick et al. have tested daily 6BN co-treatment with methadone (5–7 mg/kg, s.c.) in pregnant guinea pig dams, starting at gestational day 50 until delivery (GD~60), to prevent withdrawal behavior in guinea pig pups measured one day after birth. Even though placental 6BN transfer is slower in pregnant guinea pigs compared to mice and rhesus monkeys, the IC50 of 6BN is ~0.025 mg/kg, again displaying unexpected potency [43].

Taken together, these results demonstrate that 6BN possesses high potency in preventing the development of dependence during repeated opioid drug exposure, at doses that do not affect antinociception nor cause overt withdrawal. The high potency of 6BN in preventing dependence cannot be accounted for by classical opioid receptor models, even when 6BN access to the brain is limited. It is possible that the distribution of potent receptor ligands between blood and brain is non-linear at very low concentrations, since potent opioid antagonists tend to accumulate at the receptor and are retained in the brain—with a large portion of the total drug level in the brain bound to the receptor [45]. Such a receptor retention mechanism, assuming a discrete receptor micro-compartment where the drug is sequestered, can counteract the slow access of 6BN to the brain and enhance CNS potency for drugs with high receptor affinity.

Repeated priming with opioids can lead to hyperalgesia, at least in part mediated by peripheral afferent nociceptors [46]. Blocking MOR sites in peripheral nociceptive afferent neurons with a peripheral antagonist, methylnaltrexone, was shown to suppress development of tolerance and opioid-induced hyperalgesia (OIH) [14]. Preliminary evidence indicates that 6BN is similarly effective against OIH, again with high potency (Z. Wang, unpublished). Possibly, the same mechanisms underlie prevention of dependence and OIH with 6BN.

## 6. Hypothesis: A Novel MOR Receptor Model Relevant to Opioid Dependence Invoking a Site with High Affinity to 6BN

Our results demonstrate that 6BN prevents opioid dependence with higher potency compared to blocking antinociception or causing withdrawal. Its potency in this regard is similar to the high potency of naltrexone in blocking antinociception or causing withdrawal, whereas 6BN is two orders of magnitude less potent than naltrexone in these measures. We, therefore, propose that 6BN is a biased opioid ligand binding potently to a distinct MOR site in a non-competitive fashion and modulating dependence, expanding the concept of multiple receptor conformations with distinct ligand affinities that has enabled development of biased agonists.

How can 6BN prevent or reverse opioid dependence caused by MOR agonists? We propose a model of interacting MOR conformations that can account for the observed results with 6BN (Figure 5), in view of dynamic regulation of peripheral opioid receptors [13]. Assume a distinct MOR site (MOR-μ^x^) in equilibrium with MOR-μ, reminiscent of previously postulated ‘receptor reserve’. MOR-μ^x^ could comprise multiple receptor states, including the beta-arrestin coupled site proposed to be more prevalent in peripheral neurons [13], with each MOR conformation dependent on interacting proteins and factors in target tissues. While the proposed MOR-μ- MOR-μ^x^ equilibrium could vary as a function of cell type and could favor MOR-μ^x^ in the opioid-naïve state, in this model agonist treatment shifts the balance towards MOR-μ together with lasting enhanced spontaneous MOR-μ* activity, a hallmark of the dependent state. Assuming 6BN had high affinity for MOR-μ^x^ than for MOR-μ, higher than other opioid ligands including naltrexone, thereby stabilizing this conformation, even small doses of 6BN could reverse the MOR-μ- MOR-μ^x^ equilibrium towards the opioid-naïve state characterized by more prevalent MOR-μ^x^. In support of this hypothesis, we had observed that 6BN becomes more potent in blocking morphine’s inhibition of peristaltic motility in mice when injected before morphine, with maximum potency reached at ~100 min before morphine [40]. This long delay is not accounted for the by the rapid peak of 6BN levels reaching the circulation, but rather is consistent with gradual depletion of MOR-μ sites towards MOR-μ^x^ sites. Similarly, the ability of 6BN, but not naltrexone, to reverse elevated MOR-μ* basal activity in chronic neuropathic pain is consistent with the model’s predictions [16,44].

The nature of the postulated MOR-μ^x^ site remains elusive but parallels the two-state model proposed for MOR in peripheral afferent nociceptors [13]. Multiple forms are likely to exist, including hetero-dimeric MOR–GPCR complexes [47,48], some forms with signaling pathways opposing canonical MOR pathways. A MOR–DOR dimer was found to stimulate intracellular calcium release via Gi proteins, an effect opposing the canonical inhibition of influx calcium channels by MOR [49]. Moreover, MOR had been shown to activate calcium influx channels including TRPV1, via G proteins [50]. We had identified a MOR site in transfected HEK293 cells that stimulates calcium influx over the first 10 s of morphine exposure, followed by separate intracellular calcium release, with selectivity for epoxymorphinans (e.g., morphine, naloxone, and naltrexone) but very low affinity to other opioids (e.g., etorphine, diprenorphine, levorphanol, and fentanyl) [51]. While opposite to the well-established MOR-mediated inhibition of calcium influx channels and activation of potassium channels, this stimulatory signaling pathway is also mediated by pertussis toxin-sensitive G proteins. Its ligand binding affinities are similar to those of a labile MOR site we had identified in rat brain tissues (MOR-λ) that rapidly decays upon tissue homogenization but accounts for ~40% of all labeled ^3^H-naloxone binding sites in rat brain [52]. It is too early to speculate on the identity of the postulated MOR-μ^x^ site, whether the observed MOR-λ sites account at least in part for MOR-μ^x^, and whether it is silent or coupled to an unorthodox signaling pathway. We are now embarking on the characterization of this hypothesized high affinity 6BN site.

## 7. Potential Clinical Applications

As peripherally selective neutral opioid antagonists, 6BN and its congeners can serve as PAMORAs, treating constipation and opioid induced bowel dysfunction. Exploratory Phase I clinical trials have shown that 6BN given *ci* potently blocks morphine (10 mg/kg) induced slowing of bowel movements at doses that do not prevent opioid analgesia [40]. In a small e-IND study of methadone maintenance patients (*n* = 4), 6BN at doses up to 1mg iv caused bowel movements and limited peripheral withdrawal but no central withdrawal symptoms [53]. Its potency may be similar to that of naldemedine (0.5 mg effective dose) in treating constipation [39]. On the other hand, acting with high potency as a modulator of opioid dependence, compounds like 6BN offer multiple additional therapeutic opportunities. Among these, pharmaceutical formulations combining any opioid analgesic with low-dose 6BN could result in safer pain therapeutics, avoiding opioid dependence without precipitating withdrawal, and possibly also opioid induced hyperalgesia, an element affecting tolerance. In addition, 6BN has a longer half-life (~12 h) than typical opioid analgesics (~4 h), thereby accumulating upon frequent dosing in opioid use disorder subjects, and reaching the brain in sufficient amounts to blunt the opioid effect (Figure 6). Lastly, 6BN might facilitate opioid withdrawal under weaning protocols, followed by continued dosing at a higher dose level to prevent recidivism.

The high potency of 6BN to prevent neonatal withdrawal behavior in guinea pig pups exposed to methadone in utero [43] promises a novel preventive therapy for neonatal opioid withdrawal syndrome (NOWS), a severe form of opioid withdrawal requiring prolonged stay in neonatal intensive care units, with only palliative therapy available [54,55,56,57]. Low-dose 6BN given to pregnant women in need of opioid pain therapy or in management protocols for opioid use disorder (e.g., with methadone or buprenorphine [55,58]) has the potential to prevent NOWS without causing substantial withdrawal in both mother and fetus. Efforts are ongoing to bring 6BN into the clinic for this purpose.

Naltrexone is currently the treatment of choice for preventing relapse, but cannot be given until one to two weeks after complete weaning to avoid strong drug-induced withdrawal, which can be avoided with staggered 6BN dosing schedules. Naltrexone administration leads to higher 6BN levels compared to the parent drug, but 6BN has low potency as an antagonist against centrally mediated analgesia, and the 6BN/naltrexone ratios are quite variable between subjects—leading to the common assumption that 6BN does not contribute to naltrexone’s effects. Ultra-low naltrexone doses combined with opioid analgesics have been proposed to enhance efficacy and reduce tolerance [59,60]. However, the theoretical underpinnings for these observations remain poorly understood. Our MOR model suggests that 6BN generated as a metabolite of naltrexone can have potent effects per se, but is counteracted by naltrexone which has high affinity for MOR-μ and MOR-μ*. It is critical that these questions are resolved to enable development of optimal pain therapies and management strategies for opioid use disorder.

MOR has also been implicated in other drug use disorders, most prominently in reducing binge drinking in alcoholics [25]. While effective in a portion of subjects with alcohol use disorder, naltrexone causes aversion in some subjects leading to low compliance and cessation of therapy. It is possible that the opioid receptor is activated in alcohol use disorder, leading to elevated MOR-μ*, with naltrexone triggering opioid-like withdrawal symptoms. Selecting 6BN as an alternative to naltrexone could avoid aversive effects while maintaining efficacy.

## 8. Biased Antagonism at GPCRs and Future Studies

A review of the literature reveals overwhelming evidence towards biased agonists that engage differential receptor conformations and signaling pathways, whereas biased antagonists remain neglected [10,61]. Violin et al. [10] mention specifically the potential for both biased agonists and antagonists as it is apparent that both can bind differentially to various receptor conformations with distinct effects, but discuss only agonists in detail. Among the few specific examples of biased antagonism, a biased CCR3 antagonist was reported to prevent receptor internalization via a β-arrestin pathway while still allowing G protein coupling, thereby effectively blocking eosinophil recruitment in vivo [62]—showing that an agonist can be rendered biased by simultaneously blocking one of two pathways. Similar dual ligand effects have been reported for adrenergic [63] and dopamine receptors (aripiprazole) [61]. The opioid literature almost entirely focuses on biased agonism. Recent studies have shown that the numerous endogenous opioid peptides differ among each other in stimulating distinct signaling, as reported for opioid drugs [64,65], but all considered to act as agonists—an area worthy of further study.

The ability of low-dose 6BN selectively to block a pathway relevant to dependence adds a new dimension to biased opioid ligands. Future molecular studies need to focus on characterization of the proposed novel MOR-μ^x^ site. We have already detected longer retention of 6BN in guinea pig brain at low levels than expected from its short half-life, likely mediated by retention at the receptor with high affinity (unpublished). This finding opens an experimental approach to study properties of a MOR-μ^x^ with high affinity for 6BN.

In conclusion, we propose a novel MOR model with multiple interconverting receptor forms. Exploiting distinct ligand affinities and functions for both agonists and antagonists promises novel strategies for management of opioid use disorder and improved opioid pain therapies.

## 9. Patents

The following patents are relevant to this paper: “Combination analgesic employing opioid and neutral antagonist”. W. Sadee, E.J. Bilsky, and J. Yancey-Wrona. U.S. patent number 8,748,448 B2. Date filed November 28, 2012; and patent related patents: US6713488 B2, US8883817B2, US9061024B2, and EP2214672.

## Figures and Tables

**Figure 1 molecules-25-04163-f001:**
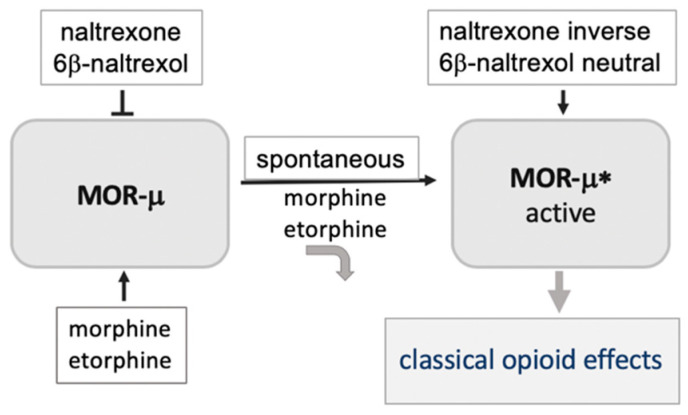
Model of the μ opioid receptor, invoking a silent ground state MOR-μ and a ligand-free activated state μ opioid receptor (MOR)-μ*. Most opioid agonists have low affinity for MOR-μ*, and therefore, dissociate from the receptor, with MOR-μ* responsible for the signaling process. The antagonists naltrexone and 6b-naltrexol (6BN) are proposed to have high affinity for both MOR-μ and MOR-μ*, blocking agonist-mediated activation of MOR-μ in a competitive fashion. Naltrexone potently blocks MOR-μ* activity as an inverse agonist, whereas the neutral antagonist 6BN binds to MOR-μ* but does not prevent signaling—both acting in a non-competitive fashion at the ligand-free MOR-μ.

**Figure 2 molecules-25-04163-f002:**
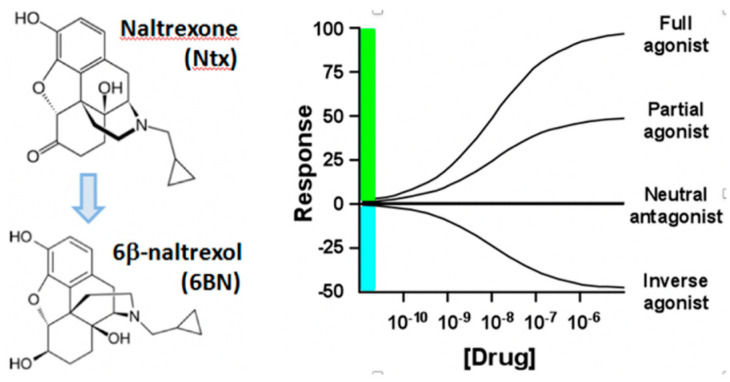
Metabolic conversion of naltrexone to 6BN, and hypothetical dose–response curves for agonists and antagonists. Etorphine is considered a full agonist and morphine a partial agonist, while 6BN is a neutral antagonist, and naltrexone an inverse agonist–the efficacy as an inverse agonist remains to be determined–measured against BTNX considered a full antagonist. Naltrexone and naloxone are near neutral antagonists in an opioid-naïve state, possibly because basal MOR-μ* activity is low in brain regions involved in withdrawal activity.

**Figure 3 molecules-25-04163-f003:**
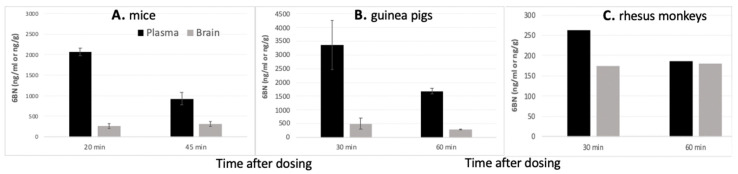
Maternal plasma and brain levels of 6BN in adult female pregnant mice, guinea pigs, and rhesus monkeys. (**A**) 6BN levels in mice at 20 and 45 min after injection (10 mg/kg, s.c.) (data from [42]). (**B**) 6BN levels in guinea pigs at 30 and 60 min after injection (10/mg/kg, s.c.) (data from [43]). (**C**) 6BN levels in rhesus monkeys at 30 and 60 min after injection (2 mg/kg, i.v.) (J. Oberdick, unpublished; n = 1 at each time point; a range of 6BN doses and regimen gave similar results).

**Figure 4 molecules-25-04163-f004:**
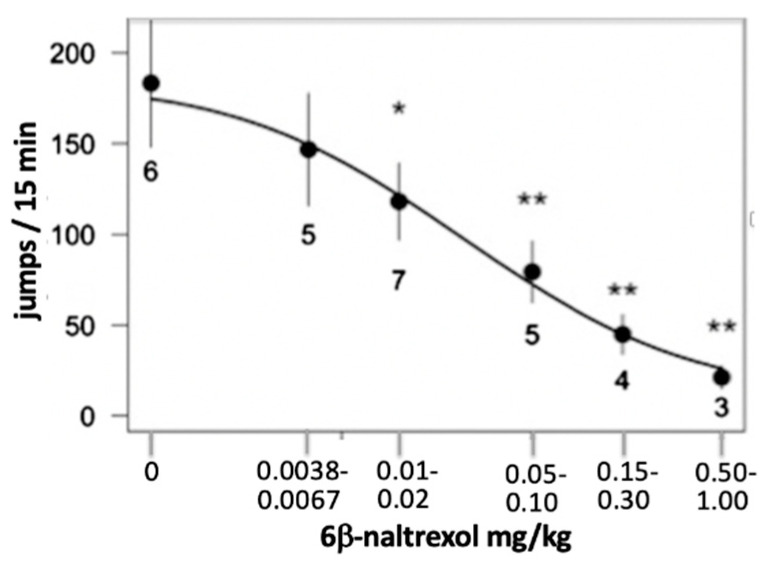
Co-administration (s.c.) of 6BN with morphine daily over 5 days to juvenile mice potently reduces naloxone-induced withdrawal behavior. Morphine injections were started on postnatal day 12 at 10 mg/kg for 3 days, followed by 3 days of 20 mg/kg. Increasing doses of 6BN were co-administered, with the dose doubled when morphine was doubled. On day 6, 30 mg/kg naloxone was injected s.c., and withdrawal jumping was measured. * *p*, 0.05; and ** *p*, 0.01 compared to no 6BN (adapted from [42]).

**Figure 5 molecules-25-04163-f005:**
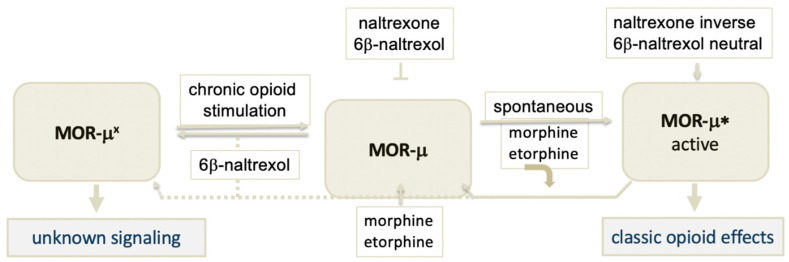
Extended model of the μ opioid receptor (MOR). Starting with the model shown in Figure 1, we add an additional receptor conformation termed MOR-μ^x^, which could exist in multiple states. We hypothesize that MOR-μ^x^ is in equilibrium with MOR-μ, and that chronic activation of MOR-μ shifts the equilibrium and depletes MOR-μ^x^, leading to elevated to MOR-μ* activity, a hallmark of the dependent state. BN is proposed to bind with high affinity to MOR-μ^x^ and stabilize this conformation, preserving the MOR-μ^x^ - MOR-μ* equilibrium of the opioid-naïve non-dependent state. It is also feasible that 6BN could facilitate conversion of MOR-μ* to MOR-μ^x^, suggested by the dotted line. This model can account for high potency of 6BN to prevent or reverse the opioid dependent state in a non-competitive fashion with opioid agonists.

**Figure 6 molecules-25-04163-f006:**
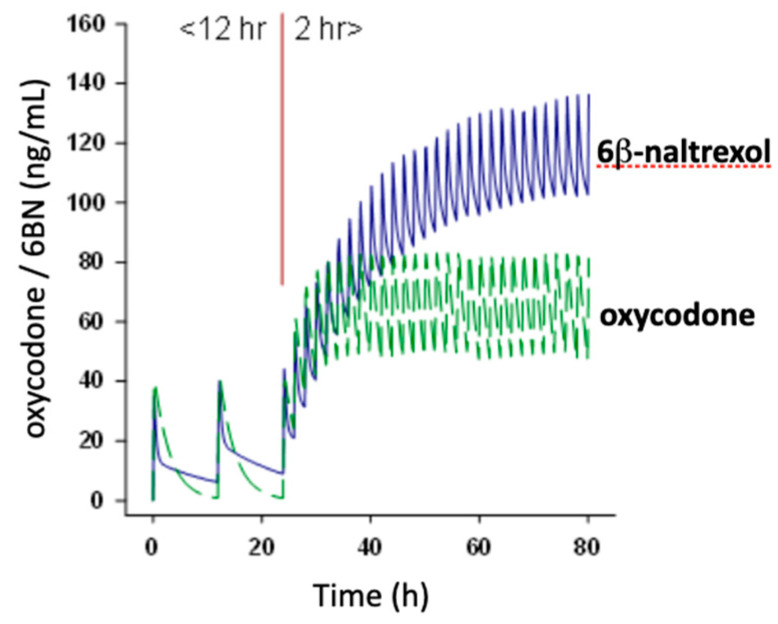
Simulated plasma levels of oxycodone (20 mg) co-administered s.c. with 6BN (15 mg). The pharmacokinetic model is based on published human blood level data for both compounds, assuming 4 and 12 h half-lives, respectively. Over the first 24 h, dosing occurs every 12 h, with low accumulation for either drug. Then, dosing continues every 2 h, simulating abuse conditions, leading to substantial accumulation of 6BN. Whereas, 15 mg 6BN is not expected to interfere with analgesia, with rapid dosing 6BN can reach levels that blunt CNS effects of the agonist, thereby reducing addiction risk.

**Table 1 molecules-25-04163-t001:** Relative potency of naltrexone and 6BN in vivo. Data are from publications that compare naltrexone with 6BN, with regards to opioid antinociception or causing withdrawal in opioid-dependent animals, and in vitro MOR binding in rhesus monkey cortical brain homogenates.

Species	Test	Agonist (Dose, Route)	Antagonist ID50, or pA2, KI Binding (Rote)		Ref.
			Naltrexone	6β-Naltrexol	
mouse	hotplate	morphine (30 mg/kg, i.p.)	0.007 mg/kg (i.p.)	1.3 mg/kg (i.p.)	[26]
mouse	withdrawal jumping	morphine (73 mg pellet, s.c., 3d)	0.09 mg/kg (i.p.)	6.9 mg/kg (i.p.)	[27]
mouse	tail-flick	hydrocodone (3.2 mg/kg, i.v.)	0.53 mg/kg (p.o.) *	2.4 mg/kg (p.o.) *	[31]
rhesus monkey	tail-withdrawal	alfentanil (0.01–5 mg/kg, s.c.)	pA2 8.5 * (0.0032–0.32 mg/kg, s.c.)	pA2 6.5* (0.32–3.2 mg/kg, s.c.)	[29]
rhesus monkey	precipitated withdrawal	morphine (6.4 mg/kg, i.m. for 3d) (respiratory functions)	0.004 mg/kg (i.m. for 3d)	0.33 mg/kg (i.m. for 3d)	[29]
rhesus monkey	MOR binding,	^3^H-DAMGO (1 nM, in vitro) ^3^H-diprenorphine (0.2 nM)	0.31 nM, Ki = 1.7 nM	Ki = 0.74 nM KI = 3.2 nM	[29]

* pA2 values are—log measures; i.e., 6BN is 100-fold less potent than naltrexone.

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
