# Peer review of "Biased Opioid Antagonists as Modulators of Opioid Dependence: Opportunities to Improve Pain Therapy and Opioid Use Management"

_molecules, 2020, doi:10.3390/molecules25184163_

Round 1

Reviewer 1 Report

Wolfgang Sadee et al. proposed biased opioid antagonists as modulators of opioid dependence with clinical application potential, and reviewed and hypothesized related pharmaceutical mechanisms.

There are some comments.

  1. The authors should carefully edit the paper throughout the manuscript before publication. For example, add the full name of 6BN in the abstract, and add the figure legends in Figure 1.
  2. For potential clinical use, I suggest the authors provide more information about the safety regarding biased opioid antagonist in the current literature.

Author Response

Reviewer 1. Comments and Suggestions for Authors

Wolfgang Sadee et al. proposed biased opioid antagonists as modulators of opioid dependence with clinical application potential, and reviewed and hypothesized related pharmaceutical mechanisms.

There are some comments.

  1. The authors should carefully edit the paper throughout the manuscript before publication. For example, add the full name of 6BN in the abstract, and add the figure legends in Figure 1.

We have added this as requested.

  1. For potential clinical use, I suggest the authors provide more information about the safety regarding biased opioid antagonist in the current literature.

There is little information on biased antagonists – we exclude allosteric ligands and bifunctional ligands at heterodimeric opioid receptors that have different efficacy at each receptor type.  We have done a literature search, but the vast majority of papers deals with biased agonist. A discussion of this literature review is presented at the end of the manuscript.  As to 6BN, our preclinical toxicology studies show it to be safe up to 300 mg/kg orally in rats.

Reviewer 2 Report

This is an excellent review of current biased opioid antagonists in opioid dependence with updated approaches for new integration, interpretation, and extension. For discussion purposes and to further increase the relevance to a broader readership in this journal (readers without strong “biased” background), this reviewer suggests a few points to consider for inclusion. I do not offer these comments recommending specific inclusion but only to encourage the authors to add a few examples to bridge your outstanding review a bit more overtly to the readers.

The authors should discuss the endogenous ligands possibly involved in this new model. Is it possible that there is a different effect of naltrexone or naltrexol on mu opioid receptors when endogenous endorphin, encephalin or dynorphin is excessively released after chronic opiate exposure, chronic pain or stress, etc? Do these environmental inputs impact on different opioid receptor statuses or confirmations in Figure 5, possibly by endogenous opioids to bind to and activate their receptors? As the expertise, the authors could provide more thoughtful discussions if there is no space limit.

Though the authors label the signal pathways with unknown in new mu model (Figure 5), it would be much better to extend it with potential future research directions if possible.

Minor points:

A number of abbreviations are used in the review, some of them are well accepted, whilst others are not. Most have been defined in the text. As a general rule, it is laborious reading an article where you have to keep flicking backwards to remind yourself of the definitions of a raft of unfamiliar abbreviations. I would recommend avoiding the excessive use of abbreviations. If the authors wish to use this style then a glossary listing all of the abbreviations is very helpful at the front of the article, which would be appreciated by the readers.

In the reference, please double check all the citations and this reviewer found that many are not matched with the ones in the text.

  • Is #10 a citation (page 10, line 416)?
  • Page 2, line 66, the citation #53 should be #13?
  • Page 5, line 213, the citation #26 should be #27?

Typos:

  • Abstract: line 26, 6BN should be 6beta-naltrexol;
  • Page 6, line 221, subtitle should be “5. 6BN…” instead of “5.6. BN…”
  • Figure 5. Neither dotted lines nor arrows are easily visible;

Author Response

Reviewer 2. Comments and Suggestions for Authors

This is an excellent review of current biased opioid antagonists in opioid dependence with updated approaches for new integration, interpretation, and extension. For discussion purposes and to further increase the relevance to a broader readership in this journal (readers without strong “biased” background), this reviewer suggests a few points to consider for inclusion. I do not offer these comments recommending specific inclusion but only to encourage the authors to add a few examples to bridge your outstanding review a bit more overtly to the readers.

The authors should discuss the endogenous ligands possibly involved in this new model. Is it possible that there is a different effect of naltrexone or naltrexol on mu opioid receptors when endogenous endorphin, encephalin or dynorphin is excessively released after chronic opiate exposure, chronic pain or stress, etc? Do these environmental inputs impact on different opioid receptor statuses or confirmations in Figure 5, possibly by endogenous opioids to bind to and activate their receptors? As the expertise, the authors could provide more thoughtful discussions if there is no space limit.

We have added a discussion at the end to address literature on biased antagonist (quite sparse compared to biased agonist, but new references are included), if one excludes allosteric ligands and bifunctional compounds that activate different receptors (eg mu plus delta).  We think that endogenous peptides will produce a similar shift towards MOR-m*, but there is evidence for biased agonist activity (now cited at the end of the text, page 10); it is clear that our receptor model can accommodate biased agonism, if for example an agonist ligand causes robust analgesia with less dependence. However, much of this will add too much speculation. 

Though the authors label the signal pathways with unknown in new mu model (Figure 5), it would be much better to extend it with potential future research directions if possible.

We have added a brief statement on future research (page 10).  Our hypothesis opens new research pathways but the main immediate issue is to define the proposed high affinity 6BN site – we have added a sentence at the end to clarify that point – new results promise a robust approach to addressing this issue.

Reviewer 3 Report

Here Sadee et al. suggested biased opioid antagonists can be used to modulate opioid dependence. As the side effects caused by morphine prevents its chronic use, efforts to develop a new method as in this study seems to be valuable.

In this study, There are some figures to help the understanding of the readers, however, I personally think that they need more explanation such as the methods used and the meaning of the results. Also, it would have been better if there was a table which could summarize the effect of naltrexone and 6B-naltrexol when used with morphine.  

1) Line 70, please specify how the peripheral and central opioid receptor interact dynamically. 

2) Line 74, 'MOR activation could then suppress pain sensation, but also lead to a vicious circle of sustained neuropathic pain' please add reference. 

3) Upon activity, 6BN may also act on other opioid receptor as mentioned by the authors. However, the explanation seems to be poor. May they also have dose-dependent effect? May they have interactions with Mu opioid receptor? 

4) What can be the supposed effect of 6BN on PAG? 

5) Morphine is known to generate various kind of side effects in different sites(e.g. itch). Could 6BN be used in this case also? 

Author Response

Reviewer 3. Comments and Suggestions for Authors

Here Sadee et al. suggested biased opioid antagonists can be used to modulate opioid dependence. As the side effects caused by morphine prevents its chronic use, efforts to develop a new method as in this study seems to be valuable.

In this study, There are some figures to help the understanding of the readers, however, I personally think that they need more explanation such as the methods used and the meaning of the results. Also, it would have been better if there was a table which could summarize the effect of naltrexone and 6B-naltrexol when used with morphine.  

We have added some more detail but many of the method are published now, specifically the mouse and guinea pig data with 6BN effects on dependence. We have added a table (Table 1) comparing 6BN and naltrexone pharmacology to clarify the important distinctions, with added discussion in the text (under “6-naltrexol (6BN).’ (paragraph starting line 190). 

1) Line 70, please specify how the peripheral and central opioid receptor interact dynamically. 

This issue has been debated in many previous papers, specifically with respect to hyperalgesia. Some argue that opioid induced hyperalgesia is mediated by peripheral nociceptors while others invoke central mechanisms – there is likely an interaction between the two. We find that 6BN potently blocks OIH in mice, presumably by peripheral action, but this is yet to be published. We have changed the sentence: “Peripheral and central opioid receptor systems could interact dynamically, for example in the induction of opioid induced hyperalgesia, reported to be mediated both centrally and peripherally (14,15).

2) Line 74, 'MOR activation could then suppress pain sensation, but also lead to a vicious circle of sustained neuropathic pain' please add reference. 

The cited references address neuropathic pain, where MOR is essential silent but gets activated by inflammatory stimuli, thereby increasing basal MOR activity which then counteracts pain sensation – yet upon prolonged activation, it appears that a vicious circle of MOR activity leads to persistent neuropathic pain.  OIH may be another example of this phenomenon.  This vicious circle is presented as a hypothesis hinted at in the cited articles; we have inserted the two references that discuss these issues (14,16, see also (44).

3) Upon activity, 6BN may also act on other opioid receptor as mentioned by the authors. However, the explanation seems to be poor. May they also have dose-dependent effect? May they have interactions with Mu opioid receptor? 

Yes, a great comment. We have studied 6BN behavior in cells transfected with MOR, DOR, and KOR, and determined antagonist activity before and after pretreatment with selective agonists for each receptor - reference (32). Each antagonist has a different profile between receptor sub-types; we have added a brief discussion on this point.  And the question of receptor heterodimers also looms large, but must be deferred to a future study. We have added: (line 209) “Similar receptor models may also apply to other opioid receptors (DOR and KOR), or even more broadly to other GPCR families. 6BN and its analogues bind to MOR, DOR, and KOR, but inverse and neutral efficacy may differ between receptors; for example, 6BN acts as an inverse agonist at KOR after agonist pretreatment in tissue culture, whereas naltrexone appears to be neutral (32). The influence of 6BN interactions with DOR and KOR remain to be studied.”

4) What can be the supposed effect of 6BN on PAG?

 Our initial hypothesis was that 6BN exerts its effect primarily at peripheral nociceptors at low doses because of partially restricted access to central neurons (but not in monkeys, CNS access unknown in humans while still being peripherally selective).  At very low doses, it now appears that 6BN slowly accumulates at central opioid receptors by high affinity binding, overcoming the access barrier.  Therefore, we think 6BN’s potent effect on dependence can be centrally mediated, but this still needs to be experimentally addressed. See also newly included future studies, where we state that 6BN appears to be retained in the adult brain at MOR sites, which could account for prolonged central effects of very low 6BN doses even in species where 6BN has slow access to the CNS.

5) Morphine is known to generate various kind of side effects in different sites(e.g. itch). Could 6BN be used in this case also? 

Yes, we think all these ADRs may also be modified by 6BN; we would like to avoid too much speculation.

Round 2

Reviewer 1 Report

The authors have addressed my major concerns.

Reviewer 3 Report

The authors have precisely answered to all my questions. 

I suggest accepting this study in the present form.